# Effects of Marine Residue-Derived Fertilizers on Strawberry Growth, Nutrient Content, Fruit Yield and Quality

Ailin Moloșag [1], Oana Cristina Pârvulescu [2,*], Violeta Alexandra Ion [1,*], Adrian Constantin Asănică [1], Raluca Soane [1], Andrei Moț [1], Aurora Dobrin [1], Mihai Frîncu [1], Anne-Kristin Løes [3], Joshua Cabell [3], Athanasios Salifoglou [4], Marios Maroulis [4], Sevasti Matsia [4], Oana Crina Bujor [1], Diana Egri [2], Tănase Dobre [2], Liliana Aurelia Bădulescu [1] and Viorica Lagunosvchi-Luchian [1]

1 Research Center for Studies of Food and Agricultural Products Quality, University of Agronomic Sciences and Veterinary Medicine of Bucharest, 59 Marasti Blvd., 011464 Bucharest, Romania; ailin.molosag.01@gmail.com (A.M.); asanica@gmail.com (A.C.A.); ralucasoane@gmail.com (R.S.); andrei.mot@qlab.usamv.ro (A.M.); aurora.dobrin@qlab.usamv.ro (A.D.); mihai.frincu@qlab.usamv.ro (M.F.); oana.bujor@qlab.usamv.ro (O.C.B.); liliana.badulescu@qlab.usamv.ro (L.A.B.); vluchian@hotmail.com (V.L.-L.)
2 Chemical and Biochemical Engineering Department, University POLITEHNICA of Bucharest, 1–7 Gheorghe Polizu Str., 011061 Bucharest, Romania; diana_egri@yahoo.com (D.E.); tghdobre@gmail.com (T.D.)
3 Norwegian Centre for Organic Agriculture, Gunnarsveg 6, NO-6630 Tingvoll, Norway; anne-kristin.loes@norsok.no (A.-K.L.); joshua.cabell@norsok.no (J.C.)
4 School of Chemical Engineering, Aristotle University of Thessaloniki, Corner of September 3rd and Egnatia St., 54124 Thessaloniki, Greece; salif@auth.gr (A.S.); info@modernanalytics.gr (M.M.); sevi.matsia@hotmail.com (S.M.)
* Correspondence: oana.parvulescu@yahoo.com (O.C.P.); violeta.ion.phd@gmail.com (V.A.I.)

**Abstract:** An outdoor experiment was performed for six months to evaluate the effects of organic fertilizers obtained from marine residual materials on strawberry plants. Three types of organic fertilizers were used, i.e., cod (*Gadus morhua*) bone powder, common ling (*Molva molva*) bone powder, and pellets obtained by mixing small cod bone powder and rockweed (*Ascophyllum nodosum*) residues. A tabletop system for strawberry cultivation was designed, in which two bare-root strawberry plants of cultivar 'Albion' were planted in a peat substrate in each pot. Five treatments were applied, i.e., cod bone powder (F1), common ling bone powder (F2), small cod bone powder and rockweed residue pellets (FA), chemical fertilizer (E), and a control (C). The number of leaves and their nutrient content, fruit yield and quality characteristics of the strawberries grown using the organic fertilizers were similar or better than those corresponding to treatments E and C. Organic fertilizers derived from the residues of fish and macroalgae could be a promising alternative to chemical fertilizers in strawberry production.

**Keywords:** algae residue; circular economy; fertilization; fish residue; soilless medium; tabletop strawberries

## 1. Introduction

Strawberries (*Fragaria × ananassa* Duch.) are among the most cultivated, appreciated, and consumed fruits in the world [1]. They contain sugars, organic acids, phenolic compounds, volatile compounds, vitamins, and minerals, which improve human health and well-being [1–8]. Ascorbic acid (vitamin C) and phenolic compounds, especially flavonoids (anthocyanins, flavonols, and flavanols) and hydrolyzable tannins (ellagitannins and gallotannins), are the main phytochemicals responsible for the antioxidant, anti-inflammatory, and anti-tumor properties of strawberries [3–11]. Moreover, various cardiovascular and neurologic benefits are associated with their consumption [3].

The consumption of strawberries and other horticultural products should increase to allow for reduced consumption of meat, according to recent sustainability recommendations. Strawberry production in Europe was 1,762,208 t (on 154,697 ha of land) in 2021,

whereas the production in Romania was 18,430 t (on 2550 ha of land) [12]. Studying the effects of relevant factors on plant growth, nutrient concentration, and fruit parameters is essential for increasing strawberry production and quality.

Plant growth and development depend on different factors, including the cultivar (cv.) type, cultivation practices, climate, and geographical origin [1,2,6–11,13–16]. Cultivation practices (e.g., open field/protected cultivation, conventional/organic/integrated system, soil/soilless technology) can significantly affect strawberry plant growth, nutrient concentration, fruit yield and quality [1,2,6,7,9–11,16–19]. Conventional cultivation is based on mineral or organo-mineral fertilizers (hereafter referred to as chemical fertilizers), which have a beneficial effect on the crop yield but can increase the occurrence of pests and disease in crops, especially when mineral nitrogen is applied in excess [20,21]. Strawberry organic cultivation relies on organic fertilizers, usually animal manure [7,16,17,19,22], vermicompost [16,19], compost [22], and plant residues (e.g., forest litter, hazelnut husk, rice hull) [19,22]. Several studies have reported that organic fertilizers have improved strawberry plant growth, commonly evaluated in terms of the root mass, leaf mass/area, and the number of leaves and runners [1,18,19,23]. Moreover, the fruit quality parameters of organically grown strawberries have been similar or better than those of conventionally grown ones [1,7,16,17,23,24]. The common quality characteristics of strawberries for consumer acceptance are appearance (color, size, and shape), firmness, taste, and aroma [4]. Sugars (fructose, glucose, and sucrose), organic acids (mainly citric acid), and phenolic compounds (predominantly anthocyanins and flavonols) give the characteristic taste of strawberry fruit, whereas volatile compounds (alcohols, aldehydes, ketones, acids, esters, and terpenes) determine its aroma [4,5,8].

Marine residual materials from the fish and seaweed industry, which are currently underutilized, should be considered for the development of organic fertilizers. Fish residues, which are rich in nitrogen (N), phosphorus (P), and calcium (Ca), can be very valuable as fertilizers [25–27]. Rosadi and Catharina [27] reported that a liquid organic fertilizer prepared from Skipjack tuna residue, which was applied at a concentration of 3.5 mL/L water, enhanced the vegetative growth of strawberry plants. Macroalgae contain N, P, Ca, potassium (K), sodium (Na), magnesium (Mg), and sulfur (S), and they are also rich in micronutrients needed by plants, especially iron (Fe), manganese (Mn), and zinc (Zn) [18]. A *Gelidium sesquipedale* red seaweed byproduct obtained from agar-agar production, which was applied at rates of 6 t/ha and 18 t/ha, had a beneficial effect on strawberry (cv. 'Fortuna') growth and development [18]. Al-Shatri et al. [23] reported that seaweed extracts (Alga 600) at concentrations of 2–8 g/L resulted in a significant increase in the fruit yield and quality of strawberries (cv. 'Albion'). In addition, macroalgae-derived materials can be used as biostimulants, increasing plant resistance to abiotic and biotic stress [28–31].

This study aimed at highlighting the effects of organic fertilizers obtained from marine residual materials, i.e., residues of fish and macroalgae, on strawberry plant growth, nutrient concentration, fruit yield and quality characteristics. The relevant parameters of leaves (their number and nutrient contents) and fruit (mass, length, width, firmness, soluble solid content, titratable acidity, and *pH*) were evaluated.

## 2. Materials and Methods

### 2.1. Plant Material

Bare-root plants of strawberry (*Fragaria × ananassa* Duch., cv. 'Albion') were supplied by Strawberry Plants SRL (Tămășeu, Romania). They were kept cold (at 1 °C) and in the dark until planting. 'Albion' is a day-neutral (everbearing) species obtained from a cross between cv. 'Diamante' and advanced selection Cal 94.16-1 [32]. It is similar to 'Diamante', but with superior quality fruit and significantly better resistance to *Phytophthora cactorum* [32]. Compared to its parent Cal 94.16-1, the fruit of cv. 'Albion' is better-flavored, firmer, and larger, and has a longer conical shape [32].

## 2.2. Fertilizers

Three types of organic fertilizers derived from marine residues were used, i.e., cod (*Gadus morhua*) bone powder (F1), common ling (*Molva molva*) bone powder (F2), and pellets obtained by mixing small cod bone powder and rockweed (*Ascophyllum nodosum*) residues (FA) (at a volumetric ratio of 9:1). The algal material, supplied by Algea AS (Kristiansund, Norway), was a residual sludge [25% dry matter (DM)] obtained after the chemical extraction of dried and ground rockweed to produce a liquid fertilizer. The rockweed was collected from wild populations along the coast of Northern Norway. Fish residual materials, provided by Bluecirc AS (Kristiansund, Norway), resulted from the processing of fish backbones from the clipfish industry by drying and sieving. Coarser particles were provided for testing in this study, whereas fine particles were used for food purposes.

The total concentrations of N in the organic fertilizers were measured based on the Dumas combustion method using an EA3100 elemental analyzer (Eurovector, Pavia, PV, Italy). The total concentrations of P, K, Ca, Na, Mg, Fe, Mn, and Zn were determined using an Agilent 7700 Series ICP-MS (Agilent Technologies, Santa Clara, CA, USA). Prior to analysis, the samples were processed in an ETHOS UP microwave digestion system (Milestone, Sorisole, BG, Italy), at a nitric acid/hydrogen peroxide ratio of 4/1 (*v/v*). All measurements were performed in triplicate.

Three commercial liquid mineral/organo-mineral fertilizers (Terrenova, Raiza Mix, and Naturmix-Mg) from Daymsa (Zaragoza, Spain) and two commercial solid mineral fertilizers (KSC II PHYT-ACTYL and KSC III PHYT-ACTYL) from Timac Agro (Bucharest, Romania) were used as chemical fertilizers. Their macronutrient and micronutrient concentrations are summarized in Table S1.

## 2.3. Strawberry Growth Experiment

An outdoor experiment was performed at the Research Center for Studies of Food Quality and Agricultural Products (University of Agronomic Sciences and Veterinary Medicine of Bucharest) for six months (6 May 2022–1 November 2022) to evaluate the effects of organic fertilizers on strawberry plant growth and development. A tabletop system for strawberry cultivation was designed (Figure 1), and its relevant characteristics were provided in our previous paper [33]. A drip irrigation system was used and set to water four times a day to facilitate plant growth under optimal conditions. A weather station (Agriculture Sentinel, Autonomous Flight Technologies, Clinceni, Romania) continuously recorded the atmospheric parameters. The monthly mean values ± standard deviations (SD) of the outdoor temperature and relative humidity as well as the monthly values of liquid precipitation (calculated by summing the daily precipitation values for each month) during the experiment are summarized in Table 1.

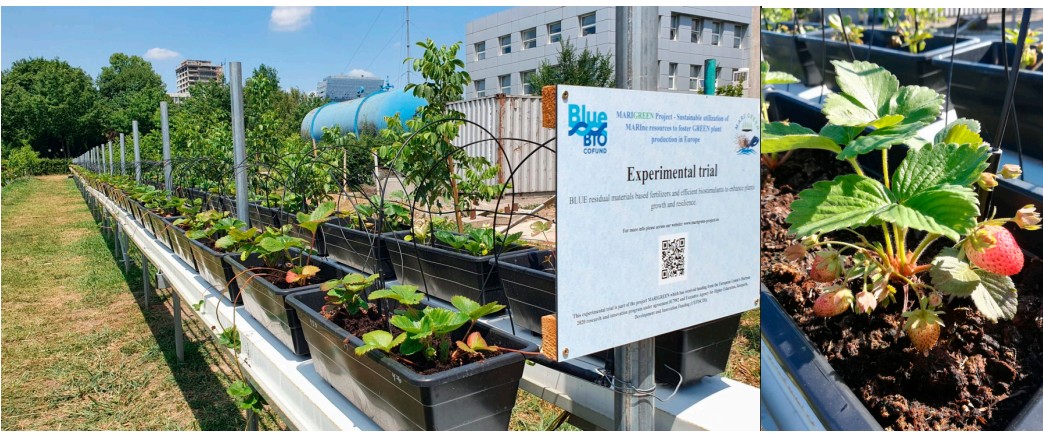

**Figure 1.** Tabletop system for strawberry cultivation.

**Table 1.** Monthly values of temperature (mean ± SD), relative humidity (mean ± SD), and liquid precipitation (sum of daily precipitation values) during the experiment.

| Month | Maximum Temperature (°C) | Minimum Temperature (°C) | Mean Temperature (°C) | Mean Relative Humidity (%) | Liquid Precipitation (L/m$^2$) |
|---|---|---|---|---|---|
| May | 23.6 ± 3.3 | 11.9 ± 2.1 | 17.6 ± 2.2 | 74.1 ± 11.0 | 59.4 |
| June | 27.0 ± 5.1 | 15.2 ± 3.2 | 20.8 ± 4.3 | 77.1 ± 8.6 | 113.8 |
| July | 32.4 ± 2.6 | 17.9 ± 1.8 | 25.2 ± 2.2 | 64.7 ± 5.7 | 27.2 |
| August | 31.6 ± 3.9 | 16.8 ± 2.6 | 24.2 ± 3.0 | 63.8 ± 8.6 | 36.8 |
| September | 25.0 ± 4.1 | 10.4 ± 2.3 | 17.5 ± 2.7 | 65.7 ± 6.6 | 5.8 |
| October | 16.0 ± 3.7 | 5.4 ± 3.1 | 10.4 ± 2.2 | 76.3 ± 11.8 | 49.8 |

Bare-root strawberries were planted in a peat substrate amended with lime and fertilizers (sphagnum peat OPM 540 W, Kekkilä-BVB, Vantaa, Finland), two plants per pot (48 cm × 18 cm × 16.3 cm). A volume of 6 L (450 g) of peat substrate was placed in each pot. The relevant physicochemical properties of the peat substrate are presented in Table S2. According to the experimental scheme shown in Table 2, five treatments were applied for strawberry growth in pots, using ten plants (5 pots) per treatment. The control treatment (C) was without fertilizer addition, except for the fertilizers initially present in the peat substrate (Table S2). The organic fertilizers corresponding to treatments F1, F2, and FA were added once a week from 19 July to 5 September 2022 (8 applications). The masses of organic fertilizers added to each application were 12 g/pot for treatment F1 and 15 g/pot for treatments F2 and FA. The organic fertilizers were mixed with 100 mL of water and 200 mL of peat and applied to the surface of the pots. The chemical fertilizers, i.e., solutions of Terrenova, Raiza Mix, and Naturmix-Mg (2 mL/L), KSC II PHYT-ACTYL (4 g/L), and KSC III PHYT-ACTYL (4 g/L), were applied to the plant roots starting from 30 June 2022. The total masses of macronutrients and micronutrients added to each plant grown with chemical fertilizer treatment are summarized in Table S3.

**Table 2.** Treatments used in the strawberry growth experiment.

| No. | Fertilizer | Code |
|---|---|---|
| 1 | - | C (control) |
| 2 | Chemical | E |
| 3 | Cod bone powder | F1 |
| 4 | Common ling bone powder | F2 |
| 5 | Fish and algae residue pellets | FA |

*2.4. Plant Analysis*

The number of plant leaves and nutrient concentrations in the leaves were evaluated six months after planting (1 November 2022). The concentrations of N and carbon (C) were measured using an EA3100 elemental analyzer (Eurovector, Pavia, PV, Italy). The concentrations of P, K, Ca, Mg, Fe, Mn, and Zn were determined similarly to those of the organic fertilizers using an ETHOS UP microwave digestion system (Milestone, Sorisole, BG, Italy) and an Agilent 7700 Series ICP-MS (Agilent Technologies, Santa Clara, CA, USA). Ten leaves per treatment (one of each plant) were selected, dried, ground, and then analyzed.

During the study, there were three fruiting periods, i.e., in June (before applying fertilizers), August, and October. Strawberry fruits were harvested at the ripe stage (bright red and dark red color). Fruit physical and chemical parameters, which were measured in each fruiting period, included the following: the mass, length, width (evaluated as the maximum diameter of the equatorial section), firmness, soluble solid content, titratable acidity, and *pH*. For each fruiting period, ten berries per treatment were analyzed.

The berries were weighed with a PS 6000.R2 precision balance (Partner Corporation, Bucharest, Romania). Their length and width were measured with a Parkside HG00962A

digital caliper (Parkside, Neckarsulm, Germany). The fruit firmness was determined using a TR 53205 digital penetrometer (T.R. TURONI, Forlì, FC, Italy) equipped with a cylindrical piston with a diameter of 8 mm. The soluble solid content was determined from the clear fruit juice, using a KRÜSS DR301-95 digital hand refractometer (A. KRÜSS Optronic, Hamburg, Germany). Titratable acidity and *pH* were measured with a TitroLine 5000 automatic titrator (SI Analytics, Mainz, Germany). Titratable acidity was determined by titrating the sample (a mixture of fruit puree and water at a mass ratio of 1/5) with 0.1 N NaOH to pH 8.1 and was expressed as g citric acid/100 g of fresh sample.

### 2.5. Statistical Analysis

One-way ANOVA with Tukey's HSD, REGWQ (Ryan, Einot, Gabriel, Welsh Studentized Range Q), and Dunett's *post hoc* tests were applied to evaluate whether the treatment and fructification period had significant effects ($p < 0.05$) on the characteristic parameters of the strawberry plant. The values of the macronutrient and micronutrient concentrations in the plant leaves were processed using principal component analysis (PCA) [1,6,8,9,19,34–38]. The Pearson correlation coefficient (*r*) was used to assess the strength of the linear correlations between different parameters. Statistical analysis was performed using XLSTAT Version 2019.1 (Addinsoft, New York, NY, USA).

## 3. Results and Discussion

### 3.1. Nutrient Concentrations in Organic Fertilizers and Nutrient Inputs per Plant

The values of the macronutrient and micronutrient concentrations in the organic fertilizers, expressed as mean ± SD, are summarized in Table 3. The mean values of the N-P-K ratios of organic fertilizers were 15-6-1 for the cod bone powder (F1), 10-9-1 for the common ling bone powder (F2), and 10-6-1 for the fish and algae pellets (FA). These were not well-balanced nutritional ratios for horticultural crops, as was the ratio in the fertilized peat substrate, i.e., 3.5-1-3.2 (Table S2). All organic fertilizers had very low concentrations of K compared to the demands of most horticultural crops.

**Table 3.** Values (mean ± SD) of nutrient concentrations in organic fertilizers.

| Element | Concentration Units | F1 | F2 | FA |
|---------|---------------------|------|------|------|
| N | g/kg | 114.5 ± 9.8 a | 89.8 ± 4.1 b | 90.1 ± 2.3 b |
| P | g/kg | 49.6 ± 1.6 c | 81.9 ± 2.2 a | 55.3 ± 1.1 b |
| K | g/kg | 7.7 ± 0.1 b | 8.7 ± 0.2 a | 9.1 ± 0.3 a |
| Ca | g/kg | 110.6 ± 4.8 c | 197.0 ± 6.7 a | 132.1 ± 3.7 b |
| Na | g/kg | 16.0 ± 0.3 b | 16.6 ± 0.3 ab | 17.0 ± 0.5 a |
| Mg | g/kg | 3.4 ± 0.1 b | 4.0 ± 0.0 a | 2.8 ± 0.0 c |
| Fe | mg/kg | 93.7 ± 0.7 b | 81.1 ± 1.5 b | 746.3 ± 39.9 a |
| Mn | mg/kg | 1.6 ± 0.0 c | 3.6 ± 0.2 b | 17.9 ± 0.4 a |
| Zn | mg/kg | 43.5 ± 0.8 b | 49.4 ± 1.4 a | 47.6 ± 0.2 a |

N: nitrogen; P: phosphorus; K: potassium; Ca: calcium; Na: sodium; Mg: magnesium; Fe: iron; Mn: manganese; Zn: zinc; F1: cod bone powder; F2: common ling bone powder; FA: fish and algae pellets. Different letters in the same row indicate a significant difference ($p < 0.05$).

The data specified in Table 3 highlight the following: (i) the mean value of the N concentration in F1 (114.5 g/kg) was significantly higher (by about 30%) than those in F2 and FA, which were similar; (ii) the mean values of the P, K, Ca, and Zn concentrations in F1 were significantly lower (by 9–78%) than those in F2 and FA; (iii) the mean value of the Na concentration in F1 (16.0 g/kg) was significantly lower (by 6%) than that in FA, but similar to that in F2 (16.6 g/kg); (iv) the mean value of the Mg concentration in F2 (4.0 g/kg) was significantly higher (up to 42%) than those in F1 (3.4 g/kg) and FA (2.8 g/kg); (v) the mean values of the Fe and Mn concentrations in FA (746.3 mg/kg and 17.9 mg/kg) were significantly higher (up to about 9 times and 11 times, respectively) than those in F1 and F2. Moreover, the mean values of the Fe concentration in the small cod fish and rockweed were 121.8 mg/kg and 431.6 mg/kg, respectively. The higher mean values of the



Fe concentration in FA (746.3 mg/kg) could have been due to Fe leakage in the pellet press, which was quite new when the pellets were produced.

The concentrations of P, Ca, Mg, and Zn, as well as those of Na, K, and Mn were directly correlated ($0.26 \leq r \leq 1.00$). Apart from the correlation between the concentrations of Mg and Zn ($r = 0.26$), the other correlations were statistically significant. Moreover, the concentrations of K and Zn were significantly directly correlated ($r = 0.73$), and they were significantly inversely correlated with the concentration of N ($r \leq -0.81$). The concentrations of Mn and Fe were significantly directly correlated ($r = 0.97$), and both were significantly inversely correlated with the concentration of Mg ($r \leq -0.82$).

Based on the mean values of the nutrient concentrations in the organic fertilizers and the fertilizer masses added to each application, the total values of the nutrient masses added to a plant during the experiment were determined (Table S3). Compared to the chemical fertilizer treatment, the nutrient inputs per plant for the organic fertilizer treatments were higher for N (up to 2.2 times), P (up to 7.2 times), and Mg (up to 8.0 times), but lower for K (up to 7.5 times), Fe (up to 4.8 times), Mn (up to 97.3 times), and Zn (up to 3.7 times). The chemical fertilizers did not contain Ca and Na. According to the results summarized in Table S3, the N-P-K ratio for the chemical fertilizer treatment was 4-1-4.

### 3.2. Characteristic Parameters of Strawberry Leaves

The mean value of the number of leaves (*NL*) per strawberry plant for treatment F1 ($\approx$16) was significantly higher (by 18–44%) than those for treatments FA and F2 ($\approx$13) and treatments C and E ($\approx$11), as shown in Figure 2. The N inputs per plant (summarized in Table S3) were 2.1–2.2 times higher for the organic fertilizer treatments (5.39–5.50 g/plant) than for the chemical fertilizer treatment (2.55 g/plant). This could be an explanation for the beneficial effects of treatments F1, F2, and FA on *NL* compared to treatments C and E.

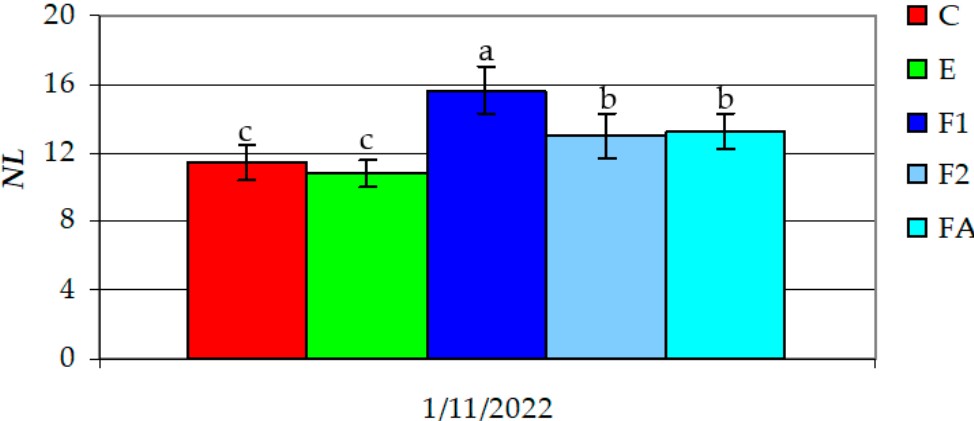

**Figure 2.** Mean values $\pm$ SD of number of leaves (*NL*) per strawberry plant for different treatments, six months after planting (1 November 2022). C: control; E: chemical fertilizers; F1: cod bone powder; F2: common ling bone powder; FA: fish and algae pellets. Different letters indicate significant differences ($p < 0.05$) among treatments.

The mean values of *NL* for treatments F1, F2, and FA were higher than those reported by Rosadi and Catharina [27] (6.2–9.7, four months after planting) using a liquid organic fertilizer prepared from Skipjack tuna residue, which was applied at concentrations of 1.5–3.5 mL/L of water. The study conducted by Al-Shatri et al. [23] for five months revealed a non-significant increase in the *NL* of strawberries cv. 'Albion' treated with seaweed extracts (Alga 600) at concentrations of 2–8 g/L (9.7–11.3) compared to the control group (9.5).

The mean values $\pm$ SD of the carbon-to-nitrogen mass ratio (*C/N*) in the plant leaves and the leaf concentrations of nitrogen (*N*), phosphorus (*P*), potassium (*K*), calcium (*Ca*), magnesium (*Mg*), iron (*Fe*), manganese (*Mn*), and zinc (*Zn*) for different treatments,

six months after planting (1 November 2022), are provided in Figure 3. The mean values of the *C/N* for the organic fertilizer treatments (17.6−18.4) were not significantly different ($p > 0.05$) and are within the range reported in other studies [39–41]. Moreover, they were significantly lower (by 26–47%) than those corresponding to the chemical fertilizer treatment (23.1) and control group (25.9), which were similar ($p > 0.05$). The mean values of *C* for all treatments (436.8–454.1 g/kg DM) were not significantly different.

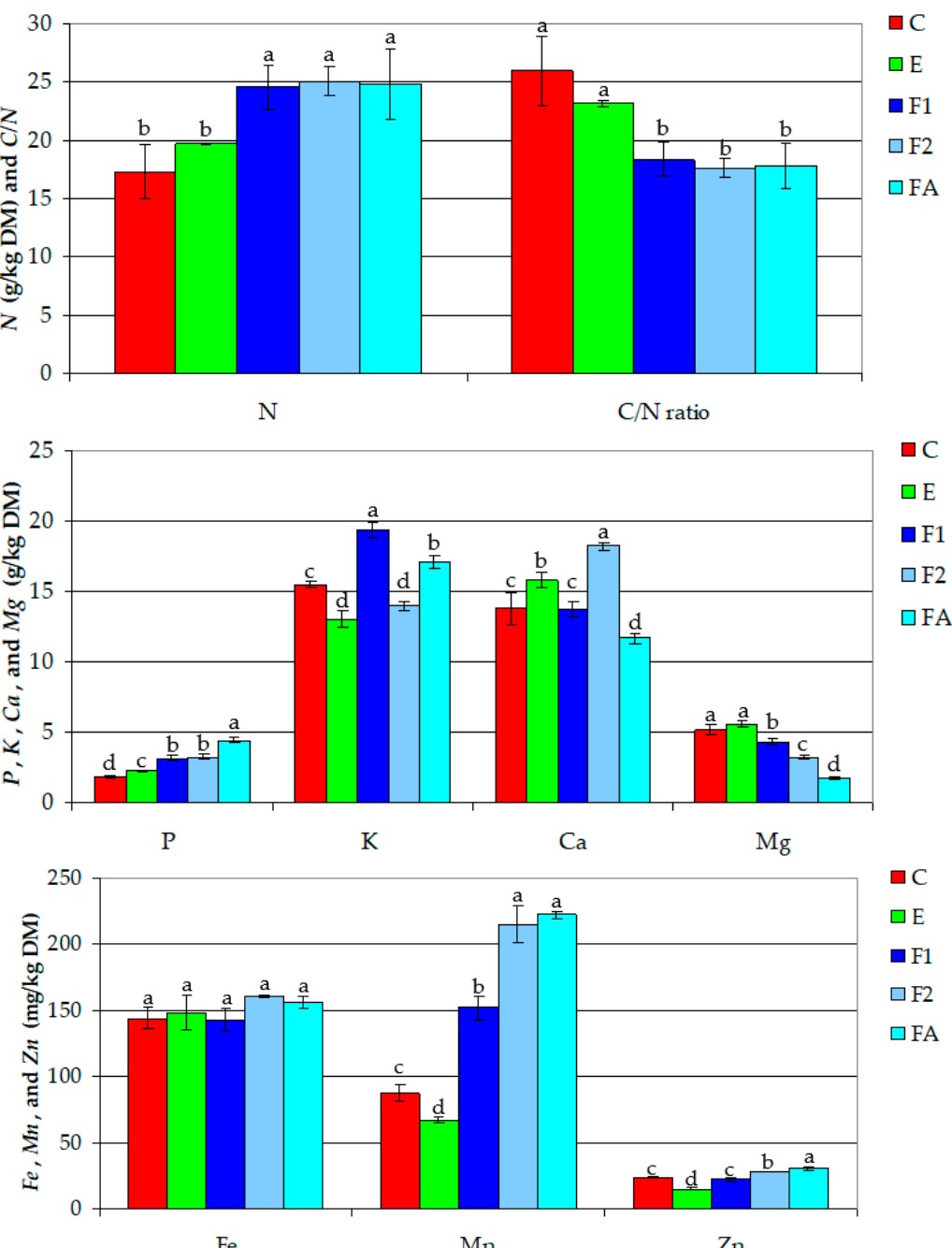

**Figure 3.** Mean values ± SD of carbon-to-nitrogen mass ratio (*C/N*) in the plant leaves and leaf concentrations of nitrogen (*N*), phosphorus (*P*), potassium (*K*), calcium (*Ca*), magnesium (*Mg*), iron (*Fe*), manganese (*Mn*), and zinc (*Zn*) for different treatments, six months after planting (1/11/2022). C: control; E: chemical fertilizers; F1: cod bone powder; F2: common ling bone powder; FA: fish and algae pellets. Different letters in the same group indicate significant differences ($p < 0.05$) among treatments.

The mean values of *N* and *P* for the organic fertilizer treatments (24.6–25.1 g/kg DM and 3.1–4.4 g/kg DM) were significantly higher (by 25–45% and 1.4–2.4 times) than those corresponding to the chemical fertilizer treatment (19.6 g/kg DM and 2.2 g/kg DM) and control group (17.3 g/kg DM and 1.9 g/kg DM). This could be due to the higher amounts of N and P added to each plant for the organic fertilizer treatments (5.39–5.50 g/plant and 2.38–4.91 g/plant) compared to the other treatments (Table S3). The mean values of *N* and *P* for treatments F1, F2, FA, and E were higher than the deficiency levels (19 g/kg DM and 2 g/kg DM), whereas those for the control group were lower [42]. The mean values of *K* for treatments F1 (19.3 g/kg DM) and FA (17.0 g/kg DM) were significantly higher (by 10–49%) than those for the other treatments (13.0–15.5 g/kg DM). The mean values of *K* for the organic fertilizer treatments (14.0–19.3 g/kg DM) and control group (15.5 g/kg DM) were higher than the deficiency level (13 g/kg DM) [42]. This indicates that the amount of K applied in the initial fertilization of the peat was enough to support the plant demand for K.

The mean value of *Ca* for treatment F2 (18.2 g/kg DM) was significantly higher (by 15–56%) than those of the other treatments (11.7–15.8 g/kg DM), probably due to the higher amount of Ca added to each plant grown with treatment F2 (11.8 g/plant) compared to the other treatments (0–7.93 g/plant). The mean values of *Ca* for all treatments (11.7–18.2 g/kg DM) were higher than the deficiency level (5 g/kg DM) [42]. Accordingly, the amount of Ca applied in the initial fertilization of the peat was sufficient to ensure the plant demand for Ca. The mean values of *Mg* for the organic fertilizer treatments (1.7–4.3 g/kg DM) were significantly lower (by 1.2–3.3 times) than those for the chemical fertilizer treatment (5.6 g/kg DM) and control group (5.2 g/kg DM). Only the mean value of *Mg* for treatment FA (1.7 g/kg DM) was lower than the deficiency level (3 g/kg DM) [42].

The mean values of *Fe* for all treatments were similar (142.5–160.6 mg/kg DM), and they were higher than the deficiency level (40 mg/kg DM) and within the optimal range of 60–250 mg/kg DM [42]. It appears that the Fe inputs per plant (4.50–21.6 mg/plant) did not affect the Fe concentration in the leaves. The mean values of *Mn* for the organic fertilizer treatments (152.6–221.9 mg/kg DM) were significantly higher (by 1.7–3.3 times) than those for the chemical fertilizer treatment (66.6 mg/kg DM) and control group (87.1 mg/kg DM). Moreover, the mean values of *Mn* for treatments F2 (214.9 mg/kg DM) and FA (221.9 mg/kg DM) were similar and significantly higher (up to 46%) than that for treatment F1. This could be due to the higher Mn inputs per plant for treatments F2 (0.22 mg/plant) and FA (1.07 mg/plant) compared to treatment F1 (0.08 mg/plant). The mean values of *Mn* for all treatments (66.6–221.9 mg/kg DM) were higher than the deficiency level (35 mg/kg DM) [42]. The mean values of *Zn* for treatments F2 (27.6 mg/kg DM) and FA (30.4 mg/kg DM) were significantly higher (by 1.2–2.0 times) than those for the other treatments (15.0–23.5 mg/kg DM). The mean values of *Zn* for all treatments (15.0–30.4 mg/kg DM) were higher than the deficiency level (10 mg/kg DM) and, except for the control group, they were within the optimal range of 20–50 mg/kg DM [42]. Consequently, in general, the plants were well supplied with micronutrients in all treatments.

The values of the macronutrient and micronutrient concentrations in the strawberry leaves (*N*, *P*, *K*, *Ca*, *Mg*, *Fe*, *Mn*, and *Zn*) were within the ranges reported in related studies [16,29,42–45]. The PCA results indicate that only the eigenvalues corresponding to PC1 (4.67) and PC2 (1.78) were > 1 and they explained 80.63% (58.38% + 22.25%) of the total variance. The data presented in Figure 4 (PCA bi-plot) and Table 4 (factor loadings) indicate that the most important variables were *Mg, Mn, P, Zn,* and *N* for PC1, and *Ca, K,* and *Fe* for PC2.

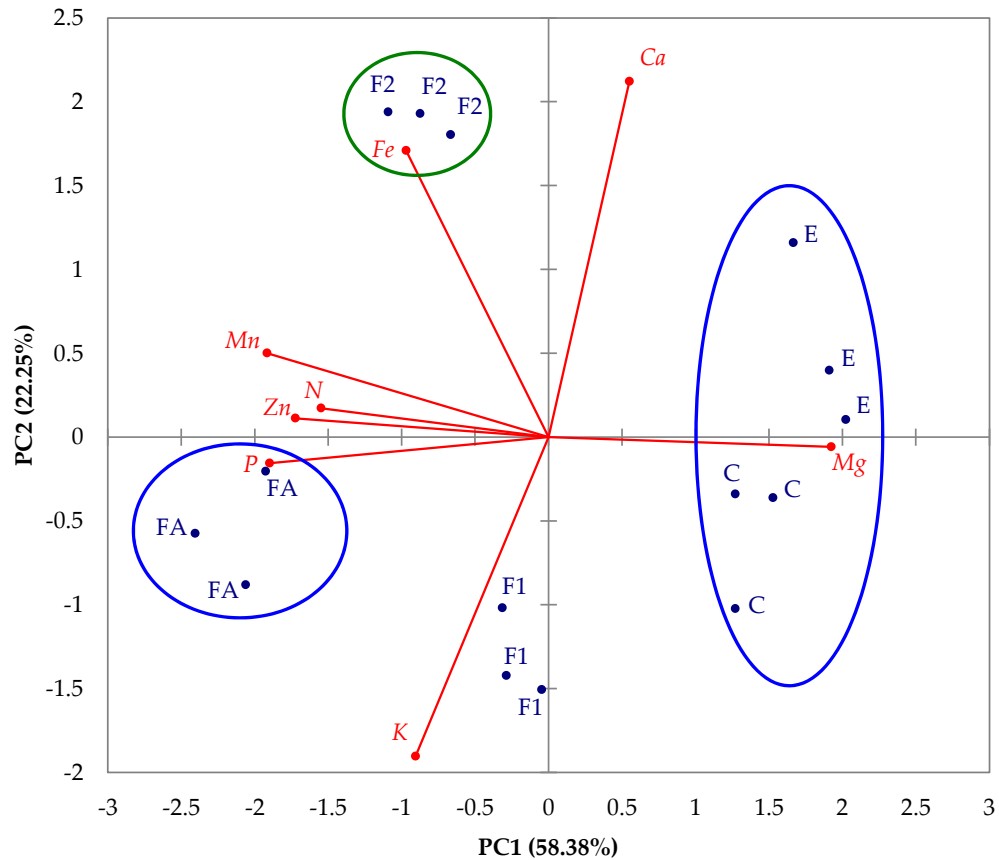

**Figure 4.** Projections of variables (*N*, *P*, *K*, *Ca*, *Mg*, *Fe*, *Mn*, and *Zn*) and samples (C, E, F1, F2, and FA) on the factor-plane PC1–PC2. *N*: leaf nitrogen concentration; *P*: leaf phosphorus concentration; *K*: leaf potassium concentration; *Ca*: leaf calcium concentration; *Mg*: leaf magnesium concentration; *Fe*: leaf iron concentration; *Mn*: leaf manganese concentration; *Zn*: leaf zinc concentration; C: control; E: chemical fertilizers; F1: cod bone powder; F2: common ling bone powder; FA: fish and algae pellets.

**Table 4.** Factor loadings.

| Variable | PC1 | PC2 |
|:---:|:---:|:---:|
| *N* | **−0.78** | 0.07 |
| *P* | **−0.96** | −0.06 |
| *K* | −0.46 | **−0.75** |
| *Ca* | 0.28 | **0.84** |
| *Mg* | **0.97** | −0.02 |
| *Fe* | −0.49 | **0.68** |
| *Mn* | **−0.96** | 0.20 |
| *Zn* | **−0.87** | 0.04 |

*N*: leaf nitrogen concentration; *P*: leaf phosphorus concentration; *K*: leaf potassium concentration; *Ca*: leaf calcium concentration; *Mg*: leaf magnesium concentration; *Fe*: leaf iron concentration; *Mn*: leaf manganese concentration; *Zn*: leaf zinc concentration; PC: principal component. Significant values of factor loadings are highlighted in bold.

The leaves of the plants grown with treatment FA had higher values of *Mn* (221.9 ± 2.9 mg/kg DM), *P* (4.4 ± 0.1 g/kg DM), *Zn* (30.4 ± 1.2 mg/kg DM), and *N* (24.8 ± 3.1 g/kg DM) but lower values of *Mg* (1.7 ± 0.1 g/kg DM) than those of the plants grown with treatments C and E (the discrimination on PC1 is highlighted in Figure 4 using blue ellipses). The results shown in Figure 3 confirm that the mean values of *Mn*, *P*, *Zn*, and *N* corresponding to treatment FA (221.9 mg/kg DM, 4.4 g/kg DM, 30.4 mg/kg DM, and 24.8 g/kg DM) were significantly higher than those corresponding to treatments C and E (66.6–87.1 mg/kg DM, 1.9–2.2 g/kg DM, 15.0–23.5 mg/kg DM, and 17.3–19.6 g/kg DM), whereas the mean value of *Mg* for treatment FA (1.7 g/kg DM) was significantly lower than

those for treatments C and E (5.2–5.6 g/kg DM). Moreover, the mean values of *Mn*, *P*, and *N* for treatments F1 and F2 (152.0–214.9 mg/kg DM, 3.1–3.2 g/kg DM, and 24.6–24.8 g/kg DM), as well as the mean value of *Zn* for treatment F2 (27.6 mg/kg DM), were significantly higher than those for treatments C and E, whereas the mean values of *Mg* for treatments F1 and F2 (3.2–4.3 g/kg DM) were significantly lower than those for treatments C and E.

The leaves of the plants grown with treatment F2 had higher values of *Ca* (18.2 ± 0.3 g/kg DM) and *Fe* (160.6 ± 1.3 mg/kg DM) but lower values of *K* (14.0 ± 0.4 g/kg DM) than those of plants grown with treatments C, F1, and FA (the discrimination on PC2 between F2, highlighted using a green ellipse, and C, F1, and FA samples can be seen in Figure 4). The results shown in Figure 3 confirm that the mean value of *Ca* in treatment F2 (18.2 g/kg DM) was significantly higher than those in treatments C, F1, and FA (11.7–13.8 g/kg DM), whereas the mean value of *K* in treatment F2 (14.0 g/kg DM) was significantly lower than those in treatments C, F1, and FA (15.5–19.3 g/kg DM). The mean value of *Fe* in treatment F2 (160.6 mg/kg DM) was not significantly higher than those in treatments C, F1, and FA (142.5–156.1 mg/kg DM).

The results presented in Table 5 (correlation matrix) indicate that *N*, *P*, *Mn*, and *Zn* were directly correlated ($0.47 \leq r \leq 0.89$) and they were inversely correlated with *Mg* ($-0.94 \leq r \leq -0.71$). Apart from the correlation between *N* and *Zn* ($r = 0.47$), the other correlations were statistically significant. Moreover, *Fe* and *Mn* were significantly directly correlated ($r = 0.55$), whereas *Ca* and *K* were significantly inversely correlated ($r = -0.62$).

**Table 5.** Correlation matrix.

| Variable | N | P | K | Ca | Mg | Fe | Mn | Zn |
|---|---|---|---|---|---|---|---|---|
| *N* | **1** | | | | | | | |
| *P* | **0.77** | **1** | | | | | | |
| *K* | 0.35 | 0.44 | **1** | | | | | |
| *Ca* | 0.01 | −0.35 | **−0.62** | **1** | | | | |
| *Mg* | **−0.71** | **−0.94** | −0.32 | 0.31 | **1** | | | |
| *Fe* | 0.24 | 0.43 | −0.22 | 0.28 | −0.45 | **1** | | |
| *Mn* | **0.78** | **0.89** | 0.33 | −0.03 | **−0.92** | **0.55** | **1** | |
| *Zn* | 0.47 | **0.73** | 0.34 | −0.23 | **−0.87** | 0.45 | **0.87** | **1** |

*N*: leaf nitrogen concentration; *P*: leaf phosphorus concentration; *K*: leaf potassium concentration; *Ca*: leaf calcium concentration; *Mg*: leaf magnesium concentration; *Fe*: leaf iron concentration; *Mn*: leaf manganese concentration; *Zn*: leaf zinc concentration. Significant values of correlation coefficients at a significance level $\alpha$ = 0.05 (two-tailed test) are highlighted in bold.

### 3.3. Fruit Yield and Physicochemical Characteristics

The effects of the fruiting period and treatments on the fruit yield per plant (*Y*) can be seen in Figure 5. For treatments F1 and F2, the mean values of *Y* were not significantly affected by the fruiting period, whereas for treatments C, E, and FA, the mean values of *Y* in the third fruiting period (171.4–202.2 g/plant) were significantly lower (by 13–25%) than those in the second fruiting period (200.4–228.1 g/plant). In the second fruiting period, the mean values of *Y* for all treatments (200.4–228.1 g/plant) were not significantly different. In the third fruiting period, the mean value of *Y* for treatment F1 (234.1 g/plant) was significantly higher (by 17–37%) than the mean values for the other treatments.

The results summarized in Table S3 highlight that the Na input per plant for the cod bone powder (F1) was about 30% lower than the Na inputs for the common ling bone powder (F2) and the fish and algae pellets (FA). This may be an explanation for the higher values of *Y* (in the third fruiting period) and *NL* (after the third fruiting period) for the plants grown with the F1 fertilizer compared to those grown with the other organic fertilizers, since strawberries are generally sensitive to Na.

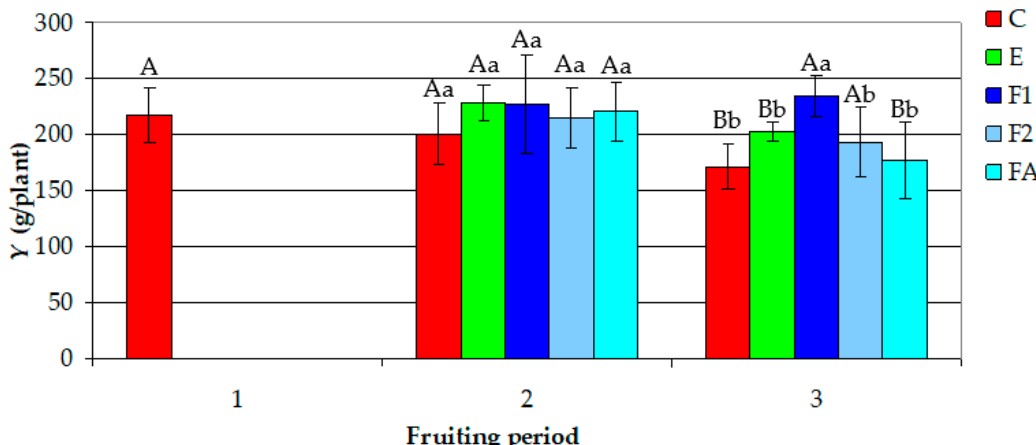

**Figure 5.** Mean values ± SD of fruit yield (*Y*); C: control; E: chemical fertilizers; F1: cod bone powder; F2: common ling bone powder; FA: fish and algae pellets. (1) June; (2) August; (3) October. Different uppercase letters indicate significant differences (*p* < 0.05) among fruiting periods; different lowercase letters indicate significant differences (*p* < 0.05) among treatments.

The mean values of *Y* obtained in this study (171.4–234.1 g/plant) are similar to those reported by Al-Shatri et al. [23] for 'Albion' strawberries grown in soil (pot experiment in an open area) (191.7–295.0 g/plant). They are higher than those obtained by Lee et al. [46] for 'Albion' strawberries grown in a hydroponic system (125.0–159.3 g/plant), but lower than those obtained by Sayğı [16] for 'Albion' strawberries grown in soil (open-field experiment using black polyethylene mulch) (272.2–408.0 g/plant).

The mean values ± SD of the fruit physical parameters for the different treatments and fruiting periods are shown in Figure 6. Except for treatment E, the fruiting period did not significantly affect the mean values of the fruit mass (*m*) or width (*d*). For treatment E, the mean values of *m* and *d* in the second fruiting period (13.9 g and 28.9 mm) were significantly higher (by 30% and 15%) than those in the third fruiting period (10.7 g and 25.1 mm). In the second and third fruiting periods, the mean values of *m* and *d* for the organic fertilizer treatments were similar (7.5–14.8 g and 22.8–27.1 mm) and not significantly different from those obtained for treatments C and E. In the second fruiting period, the mean values of *m* and *d* for treatment E were significantly higher (by 53% and 23%) than those for treatment C (9.1 g and 23.5 mm).

There were no significant differences among the mean values of fruit length (*L*) corresponding to each treatment in the three fruiting periods. In the second and third fruiting periods, the mean values of *L* for the organic fertilizer treatments were similar (27.6–36.7 mm) and not significantly different from those obtained for treatments C and E. In the second fruiting period, the mean value of *L* for treatment E (34.0 mm) was significantly higher (by 31%) than that for treatment C (26.0 mm).

Except for treatment E, the fruiting period did not significantly affect the mean values of fruit firmness (*FF*). For treatment E, the mean value of *FF* in the second fruiting period (0.68 kgf/cm$^2$) was significantly lower (by 77%) than that in the third fruiting period (1.20 kgf/cm$^2$). In the second and third fruiting periods, the mean values of *FF* for organic fertilizer treatments (0.67–1.14 kgf/cm$^2$) were similar and not significantly different from those obtained for treatments C and E.

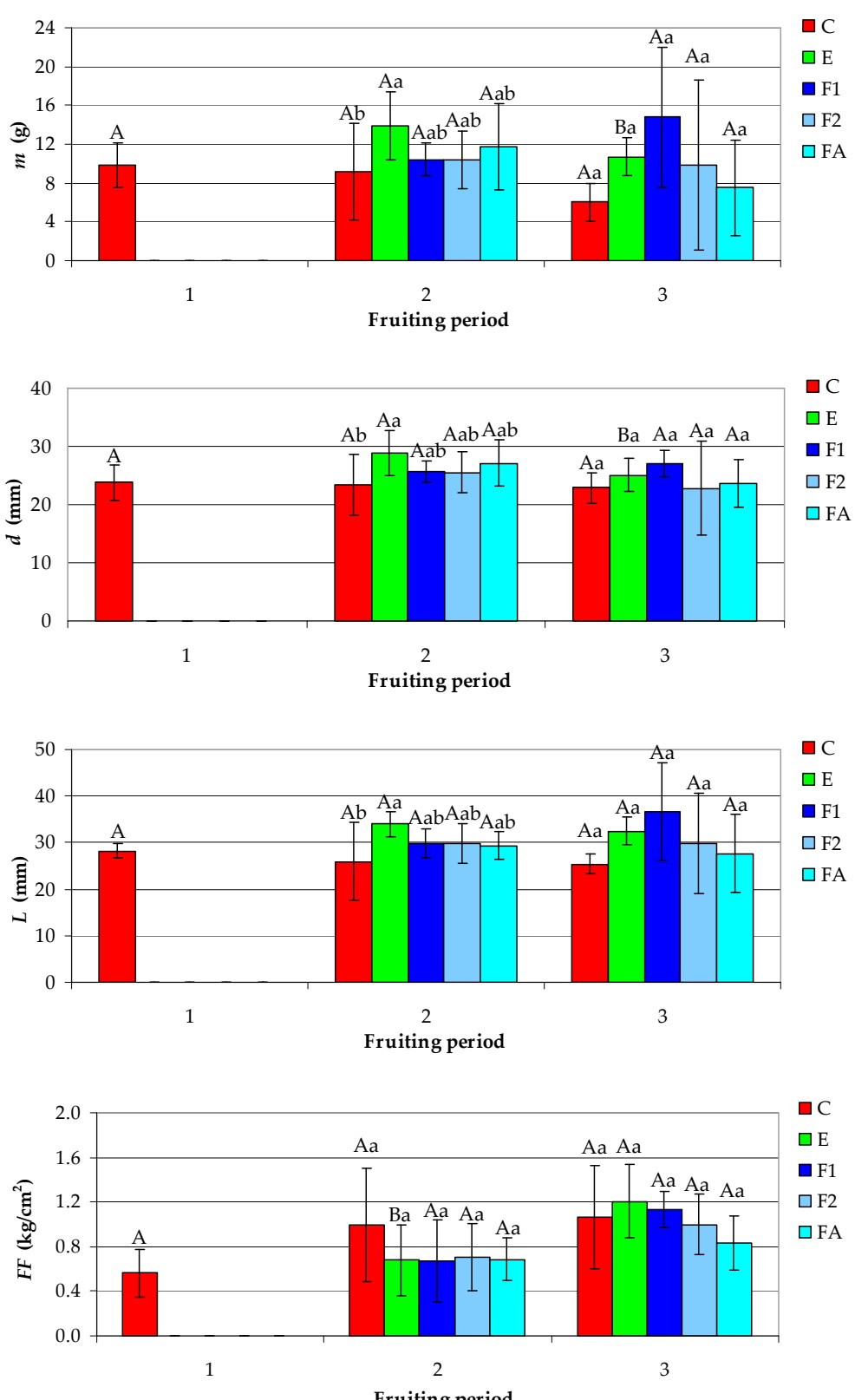

**Figure 6.** Mean values ± SD of fruit physical parameters: mass (*m*), width (*d*), length (*L*), and firmness (*FF*). C: control; E: chemical fertilizers; F1: cod bone powder; F2: common ling bone powder; FA: fish and algae pellets. (1) June; (2) August; (3) October. Different uppercase letters indicate significant differences ($p < 0.05$) among fruiting periods; different lowercase letters indicate significant differences ($p < 0.05$) among treatments.

The mean values of fruit mass ($m$), length ($L$), width ($d$), and shape index ($L/d$) obtained in this study, i.e., 6.1–14.8 g, 25.4–36.7 mm, 22.8–28.9 mm, and 1.08–1.35, are consistent with those reported in the related literature [2,15,16,23,31,47–49]. In a study on the effects of cultivar and growing location on the physicochemical properties of strawberries, Gündüz and Özbay [15] found the following mean values of $m$ for strawberries cultivated in raised beds using chemical fertilizer, in Antakya, Saksak, and Urumu (Turkey): 15.1 g (cv. 'Rubigem'), 13.9 g (cv. 'Albion'), 12.8 g (cv. 'Sweet Ann'), 12.2 g (cv. 'Fortuna'), 12.0 g (cv. 'San Andreas'), 11.1 g (cv. 'Camarosa'), 10.7 g (cv. 'Sabrina'), and 10.6 g (cv. 'Sabrosa'). Karaca and Pırlak [47] reported mean values of $m$ in the range of 5.8–8.2 g (5.8–6.3 for cv. 'Albion') for different strawberry cultivars ('Albion', 'Monterey', 'Portola', and 'San Andreas'), which were grown under ecological conditions in the Ereğli district (Konya province, Turkey). Sayğı [16] investigated the effects of different treatments, i.e., organic fertilizers (vermicompost, chicken manure, and cattle manure), chemical fertilizer, and control, on the relevant characteristics of strawberries (cv. 'Albion') cultivated in Yumurtalık (Turkey). The mean values of $m$ were in the range of 12.0–18.5 g and vermicompost had the most beneficial effect. In their study on the effect of seaweed extracts (0–8 g/L) on the physicochemical properties of strawberries (cv. 'Albion') grown in pots in an open area (Kalar, Iraq), Al-Shatri et al. [23] reported mean values of $m$ in the range of 14.7–16.9 g. In a study related to the influence of different biostimulants on the growth performance and fruit quality of strawberries (cv. 'Elsanta') grown in pots in a greenhouse (Vadena/Pfatten, Italy) using a mixture of white peat and natural clay as a growing medium, Sopelsa et al. [31] obtained fruits with mean values of $m$ between 5.7 g and 7.7 g. De Jesús Ornelas-Paz et al. [2] evaluated the physicochemical properties of organic strawberries (cv. 'Albion') grown in a field under a plastic greenhouse in Chihuahua (Mexico). The fruits were harvested at six stages of ripening (from white to dark red). The mean values of $m$ (13.0–17.6 g), $L$ (36.9–38.8 mm), $d$ (26.2–29.5 mm), and $L/d$ (1.32–1.41) did not change significantly during ripening. In their research on the growth and development of strawberries (cv. 'Albion') cultivated in pots, in a glasshouse complex (Melbourne, Australia), Balasooriya et al. [48] reported the following ranges for the mean values of $m$, $L$, $d$, and $L/d$: 4.8–12.1 g, 19–30 mm, 18–26 mm, and 1.05–1.15. Tudor et al. [49] found the following mean values of physical parameters of strawberries (cv. 'Albion') grown in a plasticulture system, in an experimental field (Bucharest, Romania), in the first year of vegetation: 8.3 g for $m$, 29.4 mm for $L$, 23.2 mm for $d$, and 1.26 for $L/d$. According to the regulations of the European Union (Commission Delegated Regulation (EU) 2019/428 of 12 July 2018 amending Implementing Regulation (EU) No 543/2011 as regards the marketing standards in the fruit and vegetables sector), marketable strawberries are classified in three classes, i.e., the "extra" class (superior quality), class 1 (good quality), and class 2 [50]. The minimum levels of the maximum diameter of the fruit equatorial section ($d$) are 25 mm for the "extra" class and 18 mm for classes 1 and 2 [50]. The values of $d$ in this study were in the range of 20–39 mm. Consequently, all fruits belonged to the "extra" class and classes 1 and 2. The percentages of the "extra" class fruits in the second and third fruiting periods were 57.1% and 66.7% for treatment F1, 50.0% and 57.1% for treatment F2, 83.3% and 40.0% for treatment FA, and 83.3% and 38.5% for treatment E, whereas those obtained for treatment C in the first, second, and third fruiting periods were in the range of 25–40%.

Fruit firmness ($FF$) is an important quality indicator, as it affects the fruit shelf life and long-distance transportability [13,16,17,51]. Gündüz and Özbay [15] reported mean values of $FF$ for different strawberry cultivars ('Albion', 'Camarosa', 'Fortuna', 'Rubigem', 'Sabrina', 'Sabrosa', 'San Andreas', and 'Sweet Ann') in the range of 0.75–0.87 kgf/cm$^2$ (0.82 kgf/cm$^2$ for cv. 'Albion'). Karaca and Pırlak [47] found mean values of $FF$ for cultivars 'Albion', 'Monterey', 'Portola', and 'San Andreas' of 1.20–1.49 kgf/cm$^2$ (1.20–1.21 for cv. 'Albion'). Sopelsa et al. [31] obtained mean values of $FF$ for strawberries (cv. 'Elsanta') between 0.58 kgf/cm$^2$ and 0.84 kgf/cm$^2$. De Jesús Ornelas-Paz et al. [2] analyzed strawberries (cv. 'Albion') harvested at six stages of ripening (from white to dark red) and found a significant decrease in the mean values of $FF$ during ripening, i.e., from 8.60 kgf/cm$^2$

to 0.77 kgf/cm$^2$. The mean values of *FF* obtained in this study (0.56–1.20 kgf/cm$^2$) are similar to those found by De Jesús Ornelas-Paz et al. [2] for bright red and dark red fruits (1.22 kgf/cm$^2$ and 0.77 kgf/cm$^2$, respectively).

The mean values $\pm$ SD of the fruit chemical parameters for the different treatments and fruiting periods are shown in Figure 7. Except for treatment FA, the fruiting period did not significantly affect the mean values of the fruit soluble solid content (*SSC*) corresponding to each treatment. For treatment FA, the mean value of the *SSC* in the third fruiting period (13.1%) was significantly higher (by 55%) than that in the second fruiting period (8.5%). In the second fruiting period, the mean values of the *SSC* for the organic fertilizer treatments were similar (8.5–13.1%) and not significantly different from those obtained for treatments C (8.9%) and E (9.3%). In the third fruiting period, the mean values of the *SSC* for treatments F2 and FA were similar (12.6% and 13.1%). They were not significantly different from those obtained for treatments F1 (11.1%) and C (11.6%) but were significantly higher (up to 24%) than the mean value corresponding to treatment E (10.6%).

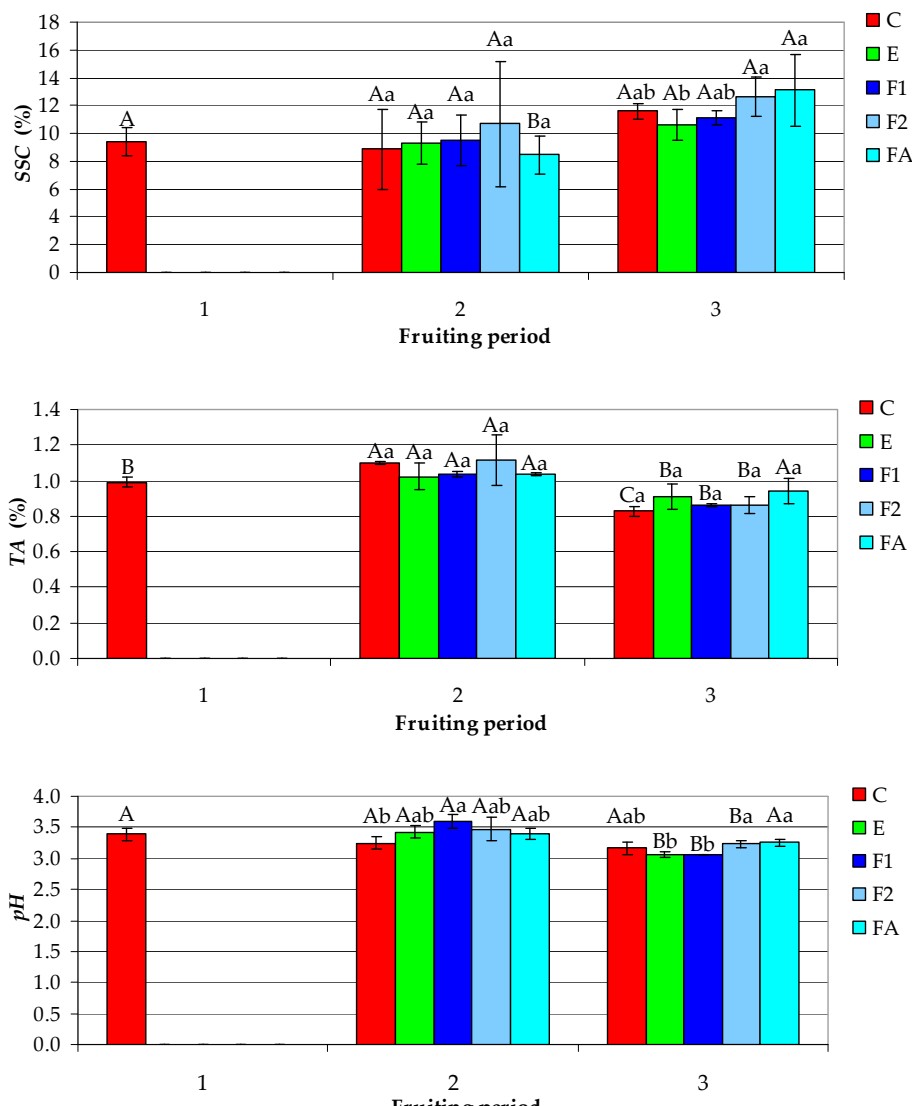

**Figure 7.** Mean values $\pm$ SD of fruit chemical parameters: soluble solid content (*SSC*), titratable acidity (*TA*), and *pH*. C: control; E: chemical fertilizers; F1: cod bone powder; F2: common ling bone powder; FA: fish and algae pellets. (1) June; (2) August; (3) October. Different uppercase letters indicate significant differences ($p < 0.05$) among fruiting periods; different lowercase letters indicate significant differences ($p < 0.05$) among treatments.

The fruiting period had a significant effect on the mean values of the fruit titratable acidity (*TA*). For all treatments, the mean values of the *TA* in the third fruiting period (0.83–0.94%) were significantly lower (by 10–33%) than those in the second fruiting period (1.02–1.12%). In the second and third fruiting periods, the mean values of the *TA* for the organic fertilizer treatments were similar (0.86–1.12%) and not significantly different from those obtained for treatments C and E.

The fruiting period did not significantly affect the mean values of the fruit *pH* for treatments FA and C but had a significant effect on those for the other treatments. For treatments E, F1, and F2, the mean values of the *pH* in the third fruiting period (3.1–3.2) were significantly lower (by 7–18%) than those in the second fruiting period (3.4–3.6). In the second fruiting period, the mean values of the *pH* for the organic fertilizer treatments were similar (3.4–3.6) and not significantly different from that obtained for the chemical fertilizer treatment (3.4). The mean value of the *pH* for treatment C (3.3) was significantly lower (by 11%) than that for treatment F1 (3.6) but not significantly different from those obtained for the other treatments (3.4–3.5). In the third fruiting period, the mean values of the *pH* for treatments F2 and FA were similar (3.2 and 3.3). They were not significantly different from that obtained for treatment C (3.2) but were significantly higher (by 6%) than those for treatments E and F1 (3.1).

The fruit chemical properties in terms of the *SSC*, *TA*, and *pH* are important quality factors because they affect the sweet and sour taste of the strawberry fruit, and thus, consumer demand [16]. The *SSC* is a relevant indicator of sugar content because sugars (fructose, glucose, and sucrose) represent about 80–90% of the soluble solids [5,16]. The *TA* estimates the concentration of organic acids in the berries. Citric acid is the predominant acid in strawberries, but they also contain other organic acids, e.g., malic, ascorbic, succinic, oxalic, tartaric, pyruvic, and fumaric [1,5,7,8]. The *SSC*, *TA*, and especially the *SSC/TA* ratio are indicators of fruit maturity, as the acid concentration tends to decrease during ripening, whereas the sugar content increases [2]. According to the results presented in Table 6, the mean values of the *SSC*, *TA*, and *pH* obtained in this study, i.e., 8.5–13.1%, 0.83–1.12%, and 3.1–3.4, are consistent with those reported in the literature for cv. 'Albion' [1,2,10,13,15,16,21,47]. Moreover, the *SSC* and *TA* were strongly inversely correlated ($-1.00 \leq r \leq -0.85$) and the values of *SSC/TA* were in the range of 5.4–15.9. These findings are in line with those obtained in other studies [2,10,13,15,23].

**Table 6.** Mean values of chemical parameters of strawberry cv. 'Albion' grown in different regions using various cultivation practices.

| Geographic Origin | Cultivation Practice | $SSC_m$ (%) | $TA_m$ (%) | $pH_m$ | Reference |
|---|---|---|---|---|---|
| Bucharest (Romania) | Outdoor experiment using a tabletop system; growing medium: peat; chemical and organic fertilizers | 8.5–13.1 | 0.83–1.12 | 3.1–3.6 | this study |
| Çukurova (Turkey) | Field experiment using low tunnel and black plastic mulch; growing medium: soil; chemical and organic fertilizers | 9.2–10.5 | 1.30–1.63 | - | [1] |
| Antakya, Saksak, and Urumu (Turkey) | Open-field experiment using raised beds and black polyethylene mulch; growing medium: soil; chemical fertilizer | 6.8 | 0.7 | 3.5 | [15] |
| Yumurtalık (Turkey) | Open-field experiment using black polyethylene mulch; growing medium: soil; chemical and organic fertilizers | 7.3–8.9 | 0.55–1.12 | 2.8–4.0 | [16] |
| Ereğli district of Konya province (Turkey) | Open-field experiment using black plastic mulch; growing medium: soil; organic fertilizer | 12.4–15.1 | 1.14–1.69 | 3.8 | [47] |

**Table 6.** *Cont.*

| Geographic Origin | Cultivation Practice | $SSC_m$ (%) | $TA_m$ (%) | $pH_m$ | Reference |
|---|---|---|---|---|---|
| Chihuahua (Mexico) | Greenhouse experiment; growing medium: soil; organic fertilizer | 6.6–9.0 | 0.7–1.2 | 3.4–3.8 | [2] |
| Marialva region, Paraná (Brazil) | Commercial crops; chemical and organic fertilizers | 8.3–9.9 | 0.81–0.90 | 3.1–3.4 | [10] |
| São José dos Pinhais (Brazil) | Field experiment using low tunnel; growing medium: soil; chemical fertilizer | 5.9–7.0 | 0.99–1.34 | - | [13] |
| Kalar (Iraq) | Pot experiment in open area; growing medium: soil; organic fertilizer | 6.9–7.5 | 0.69–0.88 | - | [23] |

($SSC_m$) mean value of soluble solid content; ($TA_m$) mean value of titratable acidity; ($pH_m$) mean value of *pH*.

## 4. Conclusions

Processing of fish for consumption and macroalgae to produce fertilizers or other products (e.g., alginate) generates significant amounts of residues, which are currently underutilized. These residual materials, which are rich in macronutrients and micronutrients beneficial for plant growth and development, can be very valuable as fertilizers. The effects of organic fertilizers obtained from fish and macroalgae residues on strawberry plant growth, nutrient concentration, fruit yield and quality parameters were evaluated in this study.

The peat substrate that was applied was fertilized with enough nutrients to give a significant yield of berries even in the third fruiting period, but the leaves were deficient in N and P after the third fruiting period, demonstrating the need for additional fertilization of this substrate when applied for strawberry growth in a tabletop system.

For the plants grown in peat amended with organic fertilizers, the number of leaves at the end of the experiment (six months after planting) was significantly higher than that for the control, probably due to the much higher N inputs per plant. Moreover, this suggests that the mineralization of N from the fish bones occurred over a long period. The percentages of the "extra" class fruits with the organic fertilizer treatments were higher than those for the control (25–40%). The results summarized in Table S3 highlight that the Na input per plant for the cod bone powder (F1) was about 30% lower than the Na inputs for the common ling bone powder (F2) and fish and algae pellets (FA). Higher values of fruit yield in the third fruiting period and the number of leaves for plants grown with the cod bone powder (F1) compared to those grown with the common ling bone powder (F2) and fish and algae pellets (FA) were obtained. This could have been due to the lower Na input per plant for the F1 fertilizer than for the other organic fertilizers, as strawberries are generally sensitive to Na. It is remarkable that the strawberries grew well in the presence of the organic fertilizers that were not well balanced with respect to the plant demand. The amount of K applied in the initial fertilization of the peat seems to have been sufficient to support the plant demand for K.

Moreover, the F2 and FA fertilizers increased the concentrations of some macronutrients and micronutrients in the plant leaves. The leaves of the plants grown with treatment FA had higher values of N, P, Mn, and Zn concentrations than those of the plants in the control group (C) and the plants grown with chemical fertilizer treatment (E). The leaves of the plants grown with treatment F2 had higher values of Ca and Fe concentrations than those of the plants grown with treatments C, F1, and FA. The fruit physicochemical characteristics, i.e., mass, length, width, firmness, soluble solid content, titratable acidity, and *pH*, of the plants grown applying organic fertilizers were not significantly different from those obtained with treatments E and C, indicating that the unbalanced fertilization and high N inputs did not adversely affect the fruit quality.

This study highlights that organic fertilizers derived from the residues of fish and macroalgae can partially replace the chemical fertilizers in strawberry cultivation. Straw-

berries, which are sensitive to Na and Cl, survived well with a fairly high application of marine-derived materials when grown in a peat substrate amended with lime and mineral fertilizers. Further studies will be needed to see what materials can be applied to produce organic fertilizers that are better balanced in terms of the nutritional demands of horticultural crops so that these organic fertilizers can partially replace the mineral fertilizers added in commercial peat substrates.

**Supplementary Materials:** The following supporting information can be downloaded at: https://www.mdpi.com/article/10.3390/agronomy13051221/s1, Table S1: Mean values of mass percentages of nutrients in chemical fertilizers; Table S2: Mean values of physicochemical properties of peat substrate; Table S3: Nutrient inputs per plant during the experiment.

**Author Contributions:** Conceptualization, O.C.P., V.A.I., A.-K.L. and A.C.A.; methodology, A.M. (Ailin Moloșag), O.C.P., V.A.I., D.E., J.C., A.M. (Andrei Moț), A.D., M.F., M.M. and S.M.; validation, O.C.P. and V.A.I.; formal analysis, O.C.P.; investigation, V.A.I., O.C.B., M.F., A.M. (Ailin Moloșag), A.M. (Andrei Moț), A.D., R.S. and L.A.B.; writing—original draft preparation, A.M. (Ailin Moloșag), O.C.P., V.A.I. and V.L.-L.; writing—review and editing, O.C.P. and V.A.I.; supervision, L.A.B., A.S., A.-K.L. and T.D. All authors have read and agreed to the published version of the manuscript.

**Funding:** This work is part of the project 244/2021 ERANET-BLUEBIO-MARIGREEN, which has received funding from the European Union's Horizon 2020 Research and Innovation Program under agreement 817992 and Ministry of Research, Innovation and Digitization, CNCS/CCCDI—UEFISCDI, within PNCDI III.

**Institutional Review Board Statement:** Not applicable.

**Informed Consent Statement:** Not applicable.

**Data Availability Statement:** Not applicable.

**Conflicts of Interest:** The authors declare no conflict of interest.

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
