# Peer review of "Effects of Marine Residue-Derived Fertilizers on Strawberry Growth, Nutrient Content, Fruit Yield and Quality"

_agronomy, doi:10.3390/agronomy13051221_

Round 1
Reviewer 1 Report
Dear Authors,
The manuscript entitled "Effects of Marine Residue-Derived Fertilizers on Strawberry Growth, Nutrient Content, Fruit Yield and Quality" presents an important topic related to the possibility of using organic marine fertilizers on the yield and quality parameters of strawberries. The manuscript has been carefully prepared. Below are my comments:
1). Abstract - incorrect order of presentation of individual fertilization variants. Variants (full names and abbreviations in parentheses) should be listed first, followed by those abbreviations
2). Introduction - carefully written, I have no objections
3). Materials and Methods
- 2.1. lack of short characteristics of the tested strawberry variety
- 2.3. lack of information on the composition of the substrate used for growing strawberries
- lack of information on the course of weather conditions during the experiment (distribution of precipitation and temperatures)
4). Results and discussion, Conclusions - no objections, good presentation of research results
Reviewer 2 Report
This study looked at highlighting the effects of organic fertilizers obtained from marine residual materials, i.e., residues of fish and macroalgae, on strawberry plant growth, nutrient concentration, fruit yield and quality characteristics.
1. What was the number of replications per treatment?
2. Under Results and discussion 3.1. – Why were results from this study higher than those reported by Rosadi and Catharina [27]?
3. After Figure 2, from the paragraph which states “Data presented in Figure 3, Figure 4 (PCA bi-plot), Table 3 (factor loadings), and Table 4 (correlation matrix) indicate the following aspects: the bullets are repeating the results in figures by mentioning the numbers. The authors could make the description more interesting by talking about proportional or percentage differences or changes between the numbers.
4. Please check for the above comment throughout the manuscript
5. Conclusion: What is the overall significance of the findings?
Not Applicable
Reviewer 3 Report
Dear Editor and Reviewers, The article is good and such a high potential to be published after major revision, mainly because it shows the potential of new alternative sources of fertilizers.
Introduction – The section is good.
Material and methods
- More details must be provided on the origin and preparation of the materials (fertilizers).
- How much of each fertilizer is added at the end - enter this information. In addition, due to the different amounts of nutrients, the authors need to calculate the inputs of each nutrient, to see if the effect is related to a greater input of nutrients by some source used. As can be seen by the higher amount of Ca in the F2 fertilizer, which probably increased the Ca content in the leaf as seen in Figure 3.
- Table 2 is more related to the results section, please move it and present the results clearly, so insert this section of the results and discussion
Results and Discussion
- The way of presenting the results must be changed, it is not used to present the data in the form of topics, sentences are used. Please correct this throughout the article.
Furthermore, the discussion, with the exception of the Fruit yield and physicochemical characteristics section, is poor and needs to be improved.
- The discussion paragraphs are long, could be more concise and have a literature review to establish the cause and effect relationship, mainly because in addition to nutrients, some fertilizers such as F3 may contain other molecules such as humic substances, which may have contributed to the improvement of the evaluated parameters.
Conclusion
The aim highligheted in this section is unnecessary, and should be moved to introduction section.
